# Improving Security in the Internet of Vehicles: A Blockchain-Based Data Sharing Scheme

Lianhai Wang [1,2] and Chenxi Guan [1,2,*]

1    The Key Laboratory of Computing Power Network and Information Security, Ministry of Education, Shandong Computer Science Center (National Supercomputer Center in Jinan), Qilu University of Technology (Shandong Academy of Sciences), Jinan 250353, China; wanglh@sdas.org
2    The Shandong Provincial Key Laboratory of Computer Networks, Shandong Fundamental Research Center for Computer Science, Jinan 250014, China
*    Correspondence: guanchenxi1106@163.com

**Abstract:** To ensure the aggregation of a high-quality global model during the data-sharing process in the Internet of Vehicles (IoV), current approaches primarily utilize gradient detection to mitigate malicious or low-quality parameter updates. However, deploying gradient detection in plain text neglects adequate privacy protection for vehicular data. This paper proposes the IoV-BDSS, a novel data-sharing scheme that integrates blockchain and hybrid privacy technologies to protect private data in gradient detection. This paper utilizes Euclidean distance to filter the similarity between vehicles and gradients, followed by encrypting the filtered gradients using secret sharing. Moreover, this paper evaluates the contribution and credibility of participating nodes, further ensuring the secure storage of high-quality models on the blockchain. Experimental results demonstrate that our approach achieves data sharing while preserving privacy and accuracy. It also exhibits resilience against 30% poisoning attacks, with a test error rate remaining below 0.16. Furthermore, our scheme incurs a lower computational overhead and faster inference speed, markedly reducing experimental costs by approximately 26% compared to similar methods, rendering it suitable for highly dynamic IoV systems with unstable communication.

**Keywords:** blockchain; federated learning; secret sharing; privacy-preserving; Internet of Vehicles

## 1. Introduction

The Internet of Vehicles (IoV) seamlessly integrates Vehicle-to-Vehicle (V2V), Vehicle-to-Road (V2R), and Vehicle-to-Infrastructure (V2I) communications by employing onboard equipment as the communication medium [1–3]. IoV represents a comprehensive network that facilitates wireless communication and data sharing among vehicles, individuals, and road infrastructure. Its versatile capabilities encompass traffic flow forecasting, vehicle position monitoring, dynamic data exchange, route selection and planning, and optimal driving experiences [4,5]. In the dynamic context of motion, vehicles continually generate diverse data types, including multimedia data, intricate driving trajectories, and real-time traffic information [6]. The abundance of data can result in communication congestion and potential redundancy [7]. Consequently, concerns regarding single points of failure and privacy breaches hinder users from actively participating [8]. Given these issues, current approaches to utilizing a centralized server for data collection become impractical [9,10]. Traditional machine learning approaches may not be well suited for highly dynamic IoV systems that are characterized by unstable communication [11].

Federated Learning (FL) has emerged as a privacy-preserving approach, ensuring both the availability and privacy of original data. Integrating FL with IoV offers various advantages, such as increased efficiency, improved privacy, decreased response time, and enhanced practicability [12,13]. The central server updates the global model by utilizing the collected local models, while the private data are stored locally. This approach can solve

specific security concerns linked to the transmission of original data, but it does introduce a reliance on third-party aggregators, thereby exposing itself to potential attacks, such as single points of failure, data tampering, and privacy breaches. The balance among these issues holds paramount importance in realizing a harmonious fusion of FL and IoV.

Blockchain provides decentralized and tamper-resistant functionality to IoV through various methods, such as data encryption, timestamps, and distributed consensus [14,15], effectively enhancing security and privacy within the IoV. By eliminating centralized management in traditional IoV systems, blockchain reduces reliance on cloud-based data storage and management, enabling any two entities (e.g., individuals, RSUs, and vehicles) to engage in peer-to-peer transactions, sharing, and communication. Directly maintaining vehicle services and transactions through blockchain facilitates significant reductions in operating costs and system risks. Establishing robust trust relationships among initially untrusted entities enhances the credibility of vehicle data through reliable verification, consequently reducing the possibility of fraud and false data dissemination. Nodes in the blockchain review uploaded models and securely store them in immutable ledgers, using cryptographic technology to enhance the security and privacy within this IoV system. Smart contracts facilitate the deployment and execution of predefined rules or scripts. In conclusion, integrating blockchain into an IoV system enhances the security, reliability, and credibility of data, reduces system costs, and enables rapid sharing and exchange of vehicle information among multiple entities [16,17]. This solution not only mitigates the risk of training nodes sharing malicious and redundant data but also ensures the security and quality of the model. Moreover, it achieves dynamic selection and control of blockchain nodes, resolving concerns of privacy breaches during task allocation processes. This, ultimately, enhances the throughput and efficiency of the IoV system [18,19].

Figure 1 illustrates that the integration of Federated Learning (FL) and blockchain is commonly used for data sharing in the Internet of Vehicles (IoV) [20–22]. However, there exists a potential vulnerability whereby training nodes can upload malicious models.

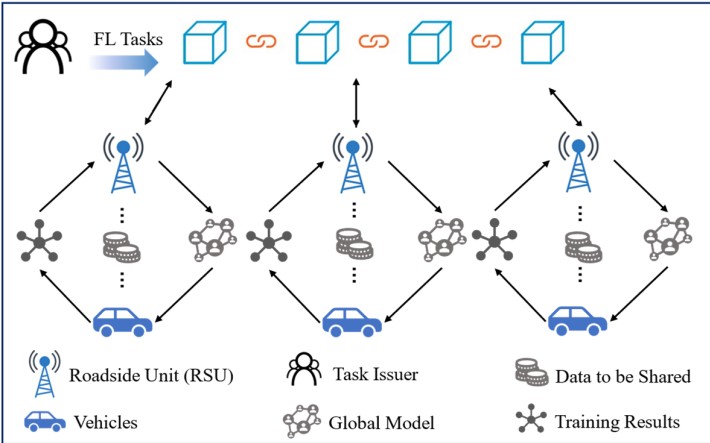

**Figure 1.** Integration of Federated Learning (FL) and blockchain in the Internet of Vehicles (IoV).

Although vehicles undergo verification upon participating in an FL task, complete trust cannot be guaranteed. Vehicles have the potential to engage in malicious behavior during the training process. If each vehicle utilizes the initial model obtained from the roadside unit, there is the possibility of intentionally tampering with a certain proportion of training results or submitting incorrect gradients [23], thereby increasing the likelihood of data misclassification and results manipulation [24]. As a result, false and invalid traffic information may manifest in the IoV system. Some scholars have used gradient detection to identify models that deviate significantly from normal models by calculating the distance. This method effectively filters out malicious model updates to prevent potential damage. However, this assumes that the central aggregator can clearly observe updates during gradient detection, without considering the potential risk of privacy breaches for

participating users. Previous studies [25] have shown that in the presence of an unsecured edge computing wireless communication channel, when a node uploads parameters to the RSU in plain text, external adversaries can still launch inference attacks on parameter updates, inferring the original training data (e.g., gradient and weight). Consequently, private information about these vehicles [26] or sensitive data on the model updates [25] (e.g., the original distribution of the training data) could be compromised. Moreover, because of the curious and dynamic nature of each vehicle, there is a possibility that it may intentionally attempt to access the private information of other vehicles. In an IoV system characterized by unstable communication, intermittent disconnections may occur, which can negatively impact the aggregation of the global model.

In this paper, we propose IoV-BDSS, a new data-sharing scheme that distinguishes itself through its expert incorporation of blockchain and federated learning techniques. This scheme enables the filtration and sharing of data within an Internet-of-Vehicles (IoV) environment. The primary contributions of this study are presented as follows:

- We designed an innovative data-sharing scheme for the IoV, employing a secret-sharing algorithm based on the Chinese remainder theorem as an alternative to homomorphic encryption. Our proposed scheme significantly mitigates the computational overhead related to encrypting and decrypting training results.
- To optimize the training effectiveness and efficiency, we innovatively utilized the Euclidean distance to select vehicles that exhibit high data similarity to the task issuer. Furthermore, we improved the Multi-Krum algorithm and integrated it with the secret-sharing algorithm to achieve ciphertext-level filtering of toxic parameter updates. This integration helps optimize the training process further.
- We implemented assessment mechanisms to evaluate the contribution of RSUs and the credibility of vehicles, further ensuring the reliability of training. Furthermore, coupled with the improved consensus mechanism, this enables the secure storage of high-quality models on the blockchain.
- Our proposed scheme was implemented utilizing FISCO BCOS as the underlying distributed framework for the IoV, coupled with FL deployment. Performance validation was conducted using two real datasets. The experimental results demonstrate its effectiveness in ensuring the protection of private data, and our model outperforms other advanced approaches in terms of the overall performance.

The following sections of this paper are organized as follows: Section 2 presents the related work; Section 3 describes the system architecture; and Section 4 introduces the overall process of the IoV-BDSS in detail. Then, Section 5 analyzes the security of the system, while Section 6 conducts simulation experiments and evaluates the performance. Finally, Section 7 presents the conclusion of the full text.

## 2. Related Work

Federated learning has shown successful applications in the Internet of Vehicles (IoV), including accurate predictions in domains like heavy vehicle driving status forecasting and unmanned drone lighting status monitoring [27]. However, the current IoV system still encounters challenges related to the privacy of vehicle information, despite notable advancements in these applications [16]. To address these challenges, the integration of blockchain and federated learning emerges as a critical component within the IoV. A Blockchain-Based Federated Learning (BFL) system was proposed by Pokhrel et al. [16], enabling updates for local on-vehicle machine learning via a consensus mechanism, eliminating the necessity of the centralized coordination of data or other entities. Chai et al. [20] introduced a federated learning framework that facilitates data sharing, leveraging a hierarchical blockchain, utilizing Roadside Units (RSUs), and implementing a two-layer Proof of Knowledge (PoK) consensus mechanism, all while formulating the transaction process as a multileader and multiparty game.

Previous schemes aimed at protecting data privacy and defending against poisoning attacks primarily utilized centralized anomaly detection, Secure Multiparty Computation

(MPC) [28], Homomorphic Encryption (HE) [29], and Differential Privacy (DP) [30]. HE guarantees that the encrypted computation results are identical to the original ones, thereby enhancing computation task security. However, because of the high computation and communication requirements and the assumption that all participating nodes are reliable [29], it is not appropriate for highly mobile IoV. In contrast, DP has lower computation and communication overheads. It adds noise to the original data, making it challenging for attackers to reverse engineer stolen results and access the original data. Additionally, DP-AFL [30] is a federated learning algorithm used for vehicular networks that incorporates Local Differential Privacy (LDP) into the training process, avoiding the security threats of centralized aggregation through a distributed asynchronous update scheme. Wang et al. [31], on the basis of Local Differential Privacy (LDP), randomized local model parameters and proposed a Reinforcement Learning (RL)-based incentive mechanism to encourage high-quality model parameters sharing by drones in dynamic systems. However, Differential Privacy (DP) disturbance is irreversible and may significantly affect training accuracy while performing poorly in Byzantine attacks. Some scholars later proposed machine learning algorithms based on secret sharing [28], which can securely output results for untrusted parties without revealing the original data in the absence of a trusted third party. The most commonly used secret-sharing scheme is Shamir [32], it has additive homomorphism, meaning that the sum of the shares of two secrets is equal to the share of the sum of the secrets, and low computational complexity, making it appropriate for the gradient aggregation process in FL. However, this scheme requires both the secret distributor and participating nodes to be honest and trustworthy.

Although the aforementioned schemes have successfully computed the aggregation results and demonstrated resilience against inference attacks, they remain susceptible to poisoning attacks. To mitigate this concern, Lu et al. [33] put forth an architecture that combines federated learning and hybrid blockchain. The objective of this architecture is to minimize transmission overhead and safeguard the privacy of data providers. They introduced a novel consensus mechanism named PoQ to mitigate the impact of malicious updates. However, identifying malicious nodes solely after the completion of training may entail supplementary computational expenses for retraining the global model, and the absence of a mechanism to eliminate these malicious nodes could result in irreversible deterioration of model accuracy. Bulyan's [34] cross-comparison scheme is well suited for nodes with independently and uniformly distributed data, although it does come with the drawback of high computational complexity. Trim Mean [35] utilizes statistical methods to compare local gradients, demonstrating effectiveness solely in scenarios in which the node data are independently and uniformly distributed. In the aggregation process, Multi-Krum [36] can identify and eliminate malevolent users, thereby ensuring that the proportion of malicious models remains within an acceptable threshold. Biscotti [37] guarantees data privacy by employing PoF consensus and Verifiable Random Functions (VRFs), while achieving secure aggregation through differential privacy and the Shamir algorithm. Nonetheless, this centralized verification method encounters challenges related to single points of failure and compromised accuracy.

There is currently a lack of a holistic and decentralized solution available to effectively mitigate inference attacks and poisoning attacks in the Internet of Vehicles (IoV). To strike a balance between development and user privacy requirements, it is crucial to conduct more comprehensive research. These efforts not only facilitate the enhancement of trust but also promote continuous technological advancements.

## 3. System Model and Security Objectives

### 3.1. System Model

Figure 2 illustrates that the IoV-BDSS consists of five main components:

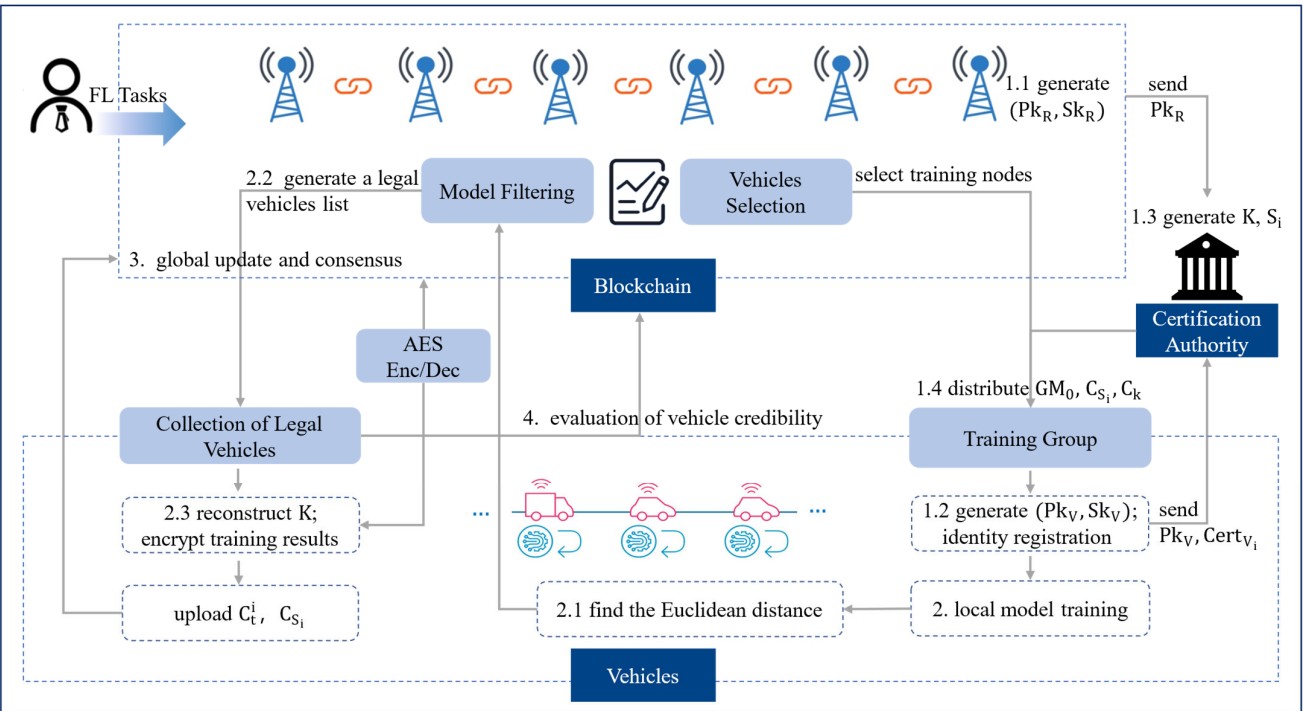

**Figure 2.** Workflow of the IoV-BDSS.

(1) Task Issuer: The task issuer creates a training model based on its needs and releases Federated Learning (FL) tasks through the blockchain. As more and more vehicles join and contribute to FL tasks, the task issuer gradually obtains an ideal model.

(2) Roadside Unit (RSU): RSUs are wireless devices installed on roadsides or specific locations to serve as crucial intermediaries between task issuers and nearby vehicles. They play a vital role in facilitating communication among these entities. RSUs are primarily responsible for distributing FL tasks, filtering out malicious parameters, decrypting, and aggregating the training models. Additionally, RSUs can operate as miner nodes within the blockchain. Multiple RSUs engage in a competitive process to validate the accuracy of the aggregated results, and the winning RSU, acting as the miner, subsequently records the validated global model on the blockchain using a consensus mechanism. Equipped with computing and storage capabilities, each RSU possesses a pair of public and private keys. Once the number of vehicle nodes uploading training results reaches the threshold of secret sharing, an RSU can restore the master key and decrypt the training results.

(3) Certification Authority (CA): The CA is responsible for generating, distributing, and maintaining secret keys, as well as registering vehicle identities. Once it has generated the master key and secret share, the CA encrypts and distributes them to the relevant parties. Vehicles participating in the FL task employ the master key provided by the CA to encrypt their training results before sending them to the RSUs. This ensures that the training data remain secure and protected during transmission and storage. In summary, the CA is responsible for managing the security of the FL process, ensuring that only authorized vehicles have access to the model and that the data are encrypted using a secure key.

(4) Vehicle: As a mobile edge computing device, the vehicle is mainly responsible for data collection, storage, and preprocessing. In each round, it downloads the global model and employs it to train local data. Additionally, it assists the RSU in the filtering process and calculates the Euclidean distance among neighboring vehicles. Then, using the master key provided by the CA, it encrypts the training results before uploading them to a nearby RSU. The training process is reiterated until the global model reaches the desired accuracy.

(5) Blockchain: The blockchain in this system is primarily constructed and maintained by distributed RSUs, which are the default configuration of the consortium chain and do

not allow external registration. These RSUs are classified into two types: miner nodes and ordinary nodes. Miner nodes possess write permissions, generate new blocks, and oversee the consensus process. Ordinary nodes do not have write permission but contribute to the system. Miner nodes are selected from ordinary nodes based on their contribution, which is evaluated using the accuracy deviation value of the aggregated results. Nodes with higher contributions will be designated as miner nodes. Consensus is reached solely among miner nodes during the verification process of a new block, without disseminating consensus messages to all ordinary nodes. Thus, this approach reduces communication overhead and enhances the consensus efficiency of the IoV system.

### 3.2. Security Objectives

(1) Ensuring the Accuracy of Global Models: To guarantee the accuracy of the global model, our proposed scheme incorporates the Multi-Krum algorithm and a consortium chain. By using a smart contract, we calculate the Euclidean distances between models and implement a secret-sharing algorithm to filter out malicious updates at the ciphertext level. Subsequently, we employ a vehicle node selection algorithm based on the Euclidean distance to choose vehicles that closely match the data of the task issuer. We evaluate the contribution of each RSU node by assessing the deviation value of accuracy within the models. RSUs that contribute more are designated as miner nodes and are responsible for storing high-quality models on the blockchain. If all participating nodes diligently adhere to the protocol process, the task issuer can, ultimately, obtain a precise model.

(2) Ensuring the Privacy of Filtered Models: Privacy is a significant concern in our scheme, especially regarding the filtered results after removing malicious parameters. To address this, we encrypt these parameter updates using a secret-sharing algorithm, ensuring that plaintext forms are not directly transmitted to RSUs. This prevents external adversaries from inferring the intermediate parameters and sensitive local data of vehicle nodes based on the ciphertext updates, thereby preserving data privacy. Importantly, both vehicles and RSU nodes will refrain from disclosing private data unless explicitly disclosing the training results. Throughout the collaborative training process, other participating vehicle nodes cannot access original data either directly or indirectly. By leveraging the threshold secret-sharing algorithm, even if a limited number of training vehicles engage in collusion, RSU nodes can algorithmically reconstruct the initial secret, providing robustness in IoV environments facing unstable communication.

## 4. Details of Our Proposed Scheme

In this section, we provide a comprehensive outline of the IoV-BDSS workflow. Table 1 below summarizes the main notations used in our proposed scheme.

**Table 1.** Summary of main notations.

| Acronym | Description |
| --- | --- |
| $N_V$ | The number of vehicles |
| $V_i$ | The i-th vehicle |
| $N_R$ | The number of RSUs |
| $R_j$ | The j-th RSU |
| $t$ | The current training round |
| $GM_0$ | The initial global model |
| $GM_t$ | The global model for the t-th round |
| $LU_i^t$ | The local update of $V_i$ for the t-th round |
| $Cert_{V_i}$ | The identity of the i-th vehicle |
| $R_a, R_b$ | The honesty and malicious thresholds |
| $Rep_{st}$ | The reputation threshold of an FL task |
| $Rep_i$ | The reputation of a vehicle |
| $M_{R_j}$ | The aggregated model of the j-th RSU |
| $Con_{R_j}^t$ | The contribution score of the j-th RSU |

### 4.1. System Initialization

(1) RSU Public–private Key Pairs Generation: The RSU randomly selects two large prime numbers, represented as p and q. It then calculates the public modulus: $m_R = p \times q$. The number of binary digits of $m_R$ is equal to the length of the RSA key $L_{RSA}$. According to the values of p, q, and $m_R$, the RSU can calculate the Euler's number, r, using the following formula: $r = \varphi(m_R) = \varphi(p) \times \varphi(q) = (p-1) \times (q-1)$. It selects an integer, $e_R$, that satisfies the condition $e_R < r$, $\gcd(e_R, r) = 1$. Next, the RSU can calculate the modular inverse element, $d_R$, such that $e_R \times d_R \equiv 1 \pmod{r}$. The j-th RSU ($j \in [1, N_R]$) possesses a public–private key pair, expressed as $\{Pk_R, Sk_R\}$. Here, $Pk_R = (m_R, e_R)$ is the public key, whereas $Sk_R = (m_R, d_R)$ is the private key. Lastly, the RSU sends the $Pk_R$ to the CA and securely stores the corresponding $Sk_R$.

(2) Vehicle Training Nodes Selection: In order to improve the efficiency and effectiveness of the training process, we propose an algorithm based on the concept of Euclidean distance. The implementation process is illustrated in Algorithm 1, which is implemented as a smart contract on the consortium chain. The algorithm aims to select vehicles with high data similarity to the task issuer, which forms a training group and assigns them FL tasks.

---

**Algorithm 1:** Vehicle Selection Contract

---

**Similarity Calculation:**

**Input:** $\text{Task} = \left\{ \text{Re}, D = \left( D_1^q, D_2^q, \cdots, D_n^q \right) \right\}$, $X_i = (X_1, X_2, \cdots, X_{N_V})$;

**Output:** $V_i = (V_1, V_2, \cdots, V_m)$;

    Initial_List[ ] = {0}, v = s = 0,

        **for** $i = 1; i < N_V + 1; i++$ **do**

            **for** $j = 1; j < n + 1; j++$ **do**

                $v = \left( D_1^q - X_{i1} \right)^2$

                $s = s + v$

            **end for**

            $\text{Similarity}(\text{issuer}, V_i) = \sqrt{s}$

            $\text{Similarity}(\text{issuer}, V_i) \rightarrow \text{Initial\_List[ ]}$

        **end for**

    Rank(Initial_List[ ])

    Select the first m vehicles from Initial_List[ ] to form $V_i = (V_1, V_2, \cdots, V_m)$

**Credibility Verification:**

**Input:** $V_i = (V_1, V_2, \cdots, V_m)$, $\text{Rep}_{st}, \text{Rep}_i$;

**Output:** TrainingGroup_List

        **for** each $V_i \in (V_1, V_2, \cdots, V_m)$ **do**

            **if** $\text{Rep}_i > \text{Rep}_{st}$

                **then** TrainingGroup_List.add($V_i$);

            **else** Remove this vehicle $V_i$

            **end if**

        **end for**

    return TrainingGroup_List

**End of Algorithm**

---

Initially, the task issuer submits an FL task to the blockchain and stipulates that the total number of vehicles in the training group is represented by m. Upon receiving this FL task, denoted as $\text{Ta} = \left\{ \text{Re}, D = \left( D_1^q, D_2^q, \cdots, D_n^q \right) \right\}$, the smart contract responsible for forming the training group is activated. Here, Re represents the specific requirements of the FL task Ta, $D = \left( D_1^q, D_2^q, \cdots, D_n^q \right)$ represents the data requirements for the vehicle training nodes, $X_i = \left( X_{i1}, X_{i2}, \cdots, X_{iN_V} \right)$ represents the local data summary of each vehicle, and

n denotes the number of feature items in the similarity calculation. The data similarity between each vehicle and the task issuer is calculated using Formula (1):

$$\text{Similarity}(\text{issuer, } V_i) = \sqrt{\left(D_1^q - X_{i1}\right)^2 + \left(D_2^q - X_{i2}\right)^2 + \cdots \left(D_n^q - X_{in}\right)^2} \quad (1)$$

Once the calculation of the data similarity between all vehicles and the task issuer is completed, the smart contract arranges the similarity values in descending order to select the top m vehicles to form a training group. To further select trustworthy vehicles for task training, Algorithm 1 also performs credibility verification on these vehicles. The details of this verification process will be introduced in the fourth part of this section. Each vehicle $V_i$ ( $i \in [1, N_m]$) also utilizes the RSA-based method to generate a public–private key pair $\{Pk_V, Sk_V\}$. The public key is $Pk_V = (m_V, e_V)$, and the private key is $Sk_V = (m_V, d_V)$.

Subsequently, $V_i$ acquires the timestamp $RT_i$ and utilizes it to sign its identity $Cert_{V_i}$, generating $Sig_{RT_i}(Cert_{V_i})$. It sends $\left\{Pk_V, Sig_{RT_i}(Cert_{V_i})\right\}$ to the CA for identity registration. Upon receiving this registration request, the CA performs initial checks on the validity of the $RT_i$ to ensure that it is within an acceptable time range. The CA then examines the validity of the $Cert_{V_i}$. Finally, the CA stores $\left\{Pk_V, Sig_{RT_i}(Cert_{V_i})\right\}$ of each vehicle in this FL task that has successfully registered. This process ensures that only authorized vehicle nodes are allowed to participate in the training task.

(3) Master Key and Secret Share Generation: The CA employs the AES key generation function to create a master key: $K = KeyGen(L_{AES})$, and $L_{AES}$ is the AES key length. It randomly chooses m mutually prime integers, $a_I = \{a_1, a_2, \cdots, a_m\}$, as the modulus and computes the Formula (2) with the assistance of K:

$$b_i = K(\text{mod } a_i) \quad (2)$$

Each vehicle's secret share is represented as $S_i = (b_i, a_i)$ in the subsequent text.

(4) Encrypted Distribution of Master Key and Secret Shares: Using the $Pk_R = (m_R, e_R)$, CA encrypts each $S_i$ resulting in the secret share ciphertext, $C_{S_i}$, obtained using Formula (3):

$$C_{S_i} = (S_i)^{e_R}(\text{mod } m_R) \quad (3)$$

Subsequently, the CA employs $Pk_V = (m_V, e_V)$ to encrypt the K, and acquires the master key ciphertext, $C_K$, using Formula (4):

$$C_K = (K)^{e_V}(\text{mod } m_V) \quad (4)$$

Finally, the CA distributes $C_{S_i}$ and $C_K$ to the vehicles in this training group, while the RSU transmits the $GM_0$ and Re to the vehicles within its coverage area.

*4.2. Local Model Training*

In the initial iteration, each vehicle receives $C_{S_i}$ and $C_K$ from the CA and simultaneously accesses the $GM_0$ and Re issued by the RSU to commence training. In the t-th iteration, each vehicle downloads the global model, $GM_{t-1}$, from the previous round for training.

(1) Find the Euclidean Distance: To address the issue of vehicles potentially uploading erroneous or low-quality parameters, we enhanced the FL by integrating Multi-Krum [36,37]. Before encrypting and uploading the training results, it enables us to find Euclidean distances between gradients, identifying gradients that deviate significantly from normal gradients as malicious. Distributing calculation tasks to each vehicle not only reduces the computational burden on the RSUs but also improves the filtering efficiency. From a security aspect, each vehicle receives a set of ciphertexts and lacks the private keys of others. This effectively prevents the explicit determination of the distances between a given

vehicle and other vehicles. Algorithm 2 illustrated the procedure for filtering out updates and generating the list of legitimate vehicles, with the detailed steps described below:

---

**Algorithm 2:** Filtering Contract

---

**Find Euclidean Distance:**
**Input:** $\text{Enc\_Pk}_V\left(\text{LU}_i^t\right)$, $\text{Enc\_Pk}_V\left(-\text{LU}_i^t\right)$, $\text{Pk}_V$
**Output:** $\text{Dis}_{\text{sum\_i}}$
    (1) **for** $j = 0$ to $N_m; j \neq i$ do
        $\text{Enc\_dis}_i[j] = \text{Enc\_Pk}_V\left(\text{LU}_i^t\right) \times \text{Enc\_Pk}_V\left(-\text{LU}_i^t\right) = \text{Enc\_Pk}_V\left(u_i^t - u_j^t\right)$;
    (2) $\text{Pk}_V \to \text{Blockchain} \to \text{Sk}_V$
    (3) **for** $j = 0$ to $N_m; j \neq i$ do
        $\text{Dis}_i[j] = \text{Dec}(\text{Enc\_dis}_i[j])$;
        $\text{Euclidean\_Dis}[j] = \|\text{Dis}_i[j]\|^2$
    (4) $\text{Euclidean\_Dis}_{\text{sum\_i}} = \sum_{j \in [0,m], j \neq i} \text{Euclidean\_Dis}_i[j]$
    (5) $\text{Sig}_{\text{Sk}_V}[\text{Euclidean\_Dis}_{\text{sum\_i}}] \& \text{Enc\_Pk}_V\left(\text{LU}_i^t\right) \to \text{RSU}$
**Parameter Filtering:**
**Input:** the set of the $\text{Euclidean\_Dis}_{\text{sum\_i}}$
**Output:** Selection\_List
    (1) $\text{SortDis} = \text{AScendSort}(\text{Euclidean\_Dis}_{\text{sum\_i}})$
    (2) $\text{Selection\_Dis} = \text{SortDis.subset}[0, m - z]$
    (3) **for** $i = 0$ to $N_m$ do
        **if** $\text{Euclidean\_Dis}_{\text{sum\_i}} \in \text{Selection\_Dis}$
          **then** Selection\_List.add$(i)$;
    **return** Selection\_List

---

In the t-th iteration, the local update of $V_i$ is denoted as $\text{LU}_i^t$. $V_i$ first encrypts $\text{LU}_i^t$ and $-\text{LU}_i^t$ using its own public key, $\text{Pk}_V$, resulting in $\text{Enc\_Pk}_V\left(\text{LU}_i^t\right)$ and $\text{Enc\_Pk}_V\left(-\text{LU}_i^t\right)$; Next, it signs $\text{Enc\_Pk}_V\left(-\text{LU}_i^t\right)$ using its own private key $\text{Sk}_V$, generating $\text{Sig}_{\text{Sk}_V} \text{Enc\_Pk}_V \left[\left(-\text{LU}_i^t\right)\right]$. The signature is shared to other vehicles in the training group for verification purposes. Upon receiving the broadcast messages from other vehicles, each vehicle node initially performs identity verification, then it uses $\text{Enc\_Pk}_V\left(\text{LU}_i^t\right)$ and $\text{Enc\_Pk}_V\left(-\text{LU}_i^t\right)$ as inputs to execute the Find Euclidean distance defined in Algorithm 2. Then, this contract sends $\text{Pk}_V$ to $V_i$ to obtain the corresponding $\text{Sk}_V$. It decrypt and find the distance. Here, m denotes the number of vehicles, z represents the number of malicious vehicles. To calculate the quality score for each local model, we sum the Euclidean distances between each vehicle's local model and its m-z-2 closest local models:

$$\text{score}(i) = \overset{\Sigma}{\underset{\substack{i \to j \\ i \neq j}}{}} \|\Delta w_i^T - \Delta w_j^T\|^2$$

where $i \to j$ signifies that the local model, $\Delta w_i^T$, is among the m-z-2 closest local models to the ideal local model, $\Delta w_j^T$. Finally, the m-z local models exhibiting the lowest quality scores are selected as the Legal\_Dis for aggregation. This filtering contract calculates $\text{Euclidean\_Dis}_{\text{sum\_i}}$, which corresponds to the total sum of Euclidean distances between the vehicle and all other vehicles. Then, $V_i$ signs the $\text{Euclidean\_Dis}_{\text{sum\_i}}$ using its own private key, and sends $\text{Sig}_{\text{Sk}_V}[\text{Euclidean\_Dis}_{\text{sum\_i}}]$, along with $\text{Enc\_Pk}_V\left(\text{LU}_i^t\right)$, to RSU.

(2) Legal Vehicle List Generation: The RSU use the output $\text{Euclidean\_Dis}_{\text{sum\_i}}$ derived from Algorithm 2. It then verifies the identity of each vehicle and chooses m-z models that are in proximity to others while disregarding the remaining models. Lastly, the filtering contract appends the identities of vehicles with retained local models to the Selection\_List set, thereby creating the final list of legal vehicles.

(3) Uploading Encrypted Training Results: the m-z vehicles identified in the legal list utilize $Sk_V = (m_V, d_V)$ to recover the K through Formula (5) using $C_K$:

$$K = (C_k)^{d_V} (\text{mod } m_V) \tag{5}$$

Lastly, $V_i$ encrypts the training result using K, resulting in $C_t^i = Enc_k\left(LU_t^i\right)$. It then sends both the ciphertext of training result $C_t^i$ and the secret share ciphertext $C_{S_i}$ to RSU.

*4.3. Global Model Update and Blockchain Consensus*

(1) Reconstruction of the Secret Share and Master Key: The RSU receives the $C_{S_i}$ and $C_t^i$ transmitted by vehicles in the legal vehicle list. Using $C_{S_i}$ and $Sk_R$, the RSU then restore the $S_i$ according to Formula (6):

$$S_i = \left(C_{S_i}\right)^{d_R} (\text{mod } m_R) \tag{6}$$

If the number of secret shares satisfies the requirements of (t, n) threshold, the RSU applies the reconstruction algorithm to restore the current round's K using more than t secret shares. If the number of secret shares is inadequate (indicating more than t vehicles have exited), the current round of reconstruction fails, and the system proceeds to the next round of global iteration. The secret reconstruction algorithm follows these steps:

First, t secret shares are selected, denoted as $S_t = \{S_{i+1}, S_{i+2}, \cdots, S_{i+t}\}$. Then, the modulus product $N_1$ is calculated using modulus $a_{i+1}$ according to Formula (7):

$$N_1 = a_{i+1} \times a_{i+2} \times \cdots \times a_{i+t} \tag{7}$$

The modulus equation set is derived using Formula (2) and denoted as Formula (8):

$$\begin{cases} K \equiv b_{i+1}(\text{mod } a_{i+1}) \\ K \equiv b_{i+2}(\text{mod } a_{i+2}) \\ \quad\vdots \\ K \equiv b_{i+t}(\text{mod } a_{i+t}) \end{cases} \tag{8}$$

The modulus equation set (8) and the modulus product $N_1$ are utilized to calculate the master key using Formula (9):

$$K = \sum_{j=1}^{i+t} b_j \times \frac{N_1}{a_j} \times \left[ \left( \frac{N_1}{a_j} \right)^{-1} \right]_{a_j} (\text{mod } N_1) \tag{9}$$

(2) Decryption and Aggregation of Training Results: The RSU decrypts the $C_t^i$ using the K and obtains the local update $LU_t^i$ for each vehicle, denoted as $LU_t^i = Dec_K\left(C_t^i\right)$.

Subsequently, utilizing all the received local training models and calculating Formula (10), the RSU can aggregate the model parameters denoted as $M_{R_j}\left(R_j \in N_R\right)$:

$$M_{R_j} = \sum_{i=1}^{M} \frac{1}{M} LU_t^i \tag{10}$$

Finally, the RSU distributes $M_{R_j}$ to all of the vehicle nodes in the current training group, thereby marking the completion of the model training.

(3) Evaluating RSU Contributions: In this section, to incentivize active participation from multiple RSUs in collaborative model training to acquire an accurate training model, we evaluate the contribution of each RSU using the accuracy deviation value. RSUs that make greater contributions are subsequently elected as miner nodes through a consensus mechanism, enabling them to record the aggregated results on the consortium chain.

Each RSU first aggregates its own model $M_{R_j}(R_j \in N_R)$ and, subsequently, transmits its model $M_{R_m}(R_m \in N_R \& m \neq j)$ to other RSU nodes while receiving the model set $\{M_{R_m}\}$ broadcasted by them. Each RSU then trains and validates $M_{R_j}$ using $\{M_{R_m}\}$, resulting in the accuracy set $\{Acc_{R_m}\}$. The accuracy deviation value is computed using Formula (11) to measure the comparative quality advantage of this RSU in relation to other RSU nodes:

$$acc_{R_j}^t = 1 + \frac{\sum_{R_m \in N_R/R_j}\left(Acc_{R_j} - Acc_{R_m}\right)}{|N_R| - 1} \tag{11}$$

By considering its historical contributions, the RSU updates its contribution score for the current round using Formula (12) based on $acc_{R_j}^t$:

$$Con_{R_j}^t = \frac{1 + acc_{R_j}^t}{1 + e^{-acc(Con_{R_j}^{t-1} \times acc_{R_j}^t)}} \tag{12}$$

(4) Blockchain Consensus Mechanism: Enhancing consensus efficiency facilitates the completion of FL tasks at a faster rate. By considering RSU's historical contributions, we incorporated a consensus based on the delegated Byzantine fault-tolerant algorithm (dBFT) [38] in this paper. The dBFT is a consensus developed based on the PBFT. In comparison to other consensuses, like PoW and DPoS, it provides the flexibility to replace miner nodes without the need for a fixed miner group and exhibits superior performance. Hence, we employ this consensus to construct a flexible consortium chain that ensures efficient and reliable federated learning. Unlike PoS, DPoS, and PBFT, which sequentially select leader nodes, the dBFT provides higher fairness by offering each node an equal opportunity to be selected. The steps of this mechanism are illustrated in Figure 3.

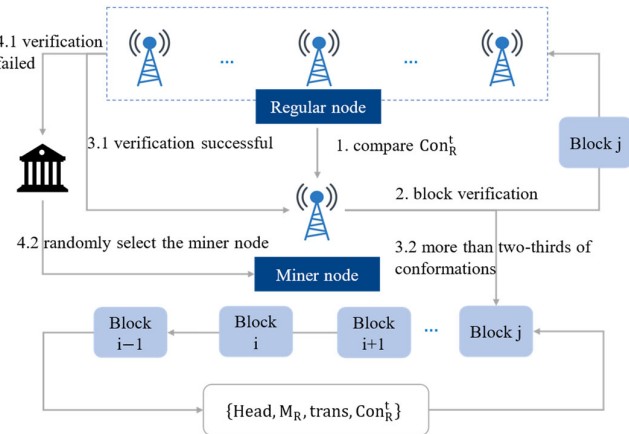

**Figure 3.** Consensus process diagram.

After the completion of the contribution calculation, the RSU with the highest score will be declared as the miner node. The miner node then generates a block encompassing global models and the contribution score for the current round. The block format adheres to the following structure:

$$Block = \left\{Head, M_{R_j}, trans, Con_{R_j}^t\right\}.$$

Here, Head represents the block header, and trans denotes the transaction information.

Following dissemination by the miner node, this new Block is then distributed to the remaining ordinary nodes for verification. The ordinary nodes validate the correctness of the Head and tra, as well as comparing their $Con_{R_j}^t$ with that of the miner node. In the case that the miner node's contribution score exceeds that of the ordinary node, the ordinary

node must acknowledge the miner node's contribution in the current round. Otherwise, the CA will select a new miner node from the pool of ordinary nodes. Once the verification is successful, the ordinary node sends a confirmation message to the miner node. When the miner node receives confirmation messages from at least two-thirds of the ordinary nodes, it confirms the legitimacy of the Block. The miner node then signs, broadcasts, and stores this new Block in the blockchain to ensure its immutability.

### 4.4. Vehicle Credibility-Based Incentive Mechanism

This paper presents an incentive mechanism based on vehicle credibility, aiming to encourage active participation and high-quality data contribution from vehicles in the training group. By integrating the Multi-Krum algorithm and reputation-based incentive protocols [39], the filtering results are leveraged to evaluate the credibility of vehicles. The credibility score can serve as an important indicator for evaluating vehicle reliability and guiding an incentive mechanism that ensures the fairness of collaborative training. Finally, the credibility score is recorded on the blockchain, enabling data tamper-proofing and nonrepudiation. The specific steps are illustrated in Figure 4.

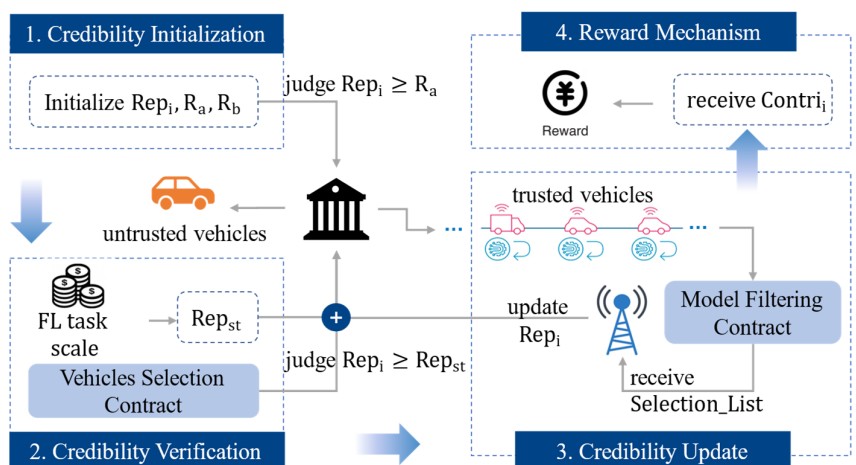

**Figure 4.** Workflow of the incentive mechanism.

(1) Credibility Initialization: Each vehicle initializes its own credibility, $Rep_i$, which is stored in the CA along with its public key and identity certificate. $Rep_i$ is an integer that belongs to the finite set $[0, R_{max}]$, where $R_a$ and $R_b$ represent the trustworthy threshold and malicious threshold within this credibility set. If a vehicle node's credibility satisfies $Rep_i \geq R_a$, it is considered trustworthy and its credibility is initialized to $R_a$. If $Rep_i \leq R_b$, the vehicle is deemed untrustworthy, and its identity certificate is revoked by the CA.

(2) Credibility Verification: The task issuer needs to assess the benefits of the global model and allocate task rewards for the training task. This approach serves to incentivize vehicles in the training group to upload high-quality parameters while also qualifying the contribution of each vehicle to the global model. Additionally, the task issuer establishes the credibility threshold, $Rep_{st}$, according to training task requirements.

Vehicles will be deemed qualified to participate in an FL task only if their credibility meets the condition $Rep_i \geq Rep_{st}$. The sensitivity of the global model being trained impacts the selection of the credible threshold: a higher threshold is required for highly sensitive global models, while a smaller threshold can be set for models with low sensitivity and a need for more data samples. During the vehicle training node selection phase, Algorithm 1 verifies that the credibility of the vehicles meets or exceeds the credibility threshold to ensure that credible vehicles participate in the training task.

(3) Credibility Update: After successfully passing the verification and joining the group, vehicles begin local training and transmit the latest models to RSUs. During each

iteration, vehicles uploading illegal parameters are recorded. Upon receiving the Selection_List from Algorithm 2, RSUs update the credibility of each vehicle using Formula (11):

$$\begin{cases} \text{Rep}_i = \text{Rep}_i + 1, \ i \in \text{Selection\_List} \\ \text{Rep}_i = \text{Rep}_i - 1, \ i \notin \text{Selection\_List} \end{cases} \tag{13}$$

Once the FL task is completed, the miner node will upload the credibility of the vehicles to the blockchain. All task issuers can select well-performing vehicles to participate in FL tasks by querying their credibility. If a vehicle's credibility falls below the set threshold, the contract will remove the vehicle from the training group.

(4) Contribution-Based Reward Mechanism: This mechanism commences by initializing the contribution score of each vehicle to 0. The contribution of the vehicle is represented by $\text{Contri}_i$, which is calculated using Formula (12):

$$\begin{cases} \text{Contri}_i = \text{Contri}_i + 1, \text{Rep}_i > \text{Rep}_{st} \\ \text{Contri}_i = \text{Contri}_i, \text{Rep}_i \leq \text{Rep}_{st} \end{cases} \tag{14}$$

The reward calculation based on $\text{Contri}_i$ is performed using Formula (13):

$$\begin{cases} \text{Reward}_i = 0, \ \text{Contri}_i = 0 \\ \text{Reward}_i = \left( \text{Contri}_i / \sum_{\text{Contri}_i \geq \frac{2t}{3}} \text{Contri}_i + 1 \right) \times \text{Reward}_i, \ \text{Contri}_i > 0 \end{cases} \tag{15}$$

## 5. Security Analysis

### 5.1. Ensuring Accuracy of the Global Model

(1) Implementation of the Multi-Krum Algorithm: One of our main strategies involves deploying the Multi-Krum algorithm, which is tightly integrated with the consortium chain via smart contracts to discern the disparities among gradients. This method enables us to detect parameters deviating from normal gradients, identifying them as malicious and excluding them. The filtering results serve as a trustworthiness metric for evaluating the behavior of vehicles involved in the collaborative training process. This enhancement of trust proves indispensable in the subsequent training processes, facilitating the selection of exceptionally reliable vehicles for participation.

(2) Node Selection Algorithm Based on Euclidean Distance: Leveraging the principles of Euclidean distance, we designed a vehicle selection algorithm to optimize training efficiency. This selection algorithm proficiently selects a few highly matching vehicles that exhibit significant compatibility. Subsequently, this process further ensures the efficient acquisition of an accurate global model within our system.

(3) Contribution Assessment and Assignment of Miner Nodes: We assess the contribution scores of all RSUs based on the accuracy deviation value. The RSU nodes that demonstrate substantial scores are designated as miner nodes, responsible for writing high-quality models to the blockchain. This methodology comprehensively evaluates the contribution of the RSUs, ensuring the correctness and high quality of the final aggregation results.

### 5.2. Protecting the Privacy of Filtering Results

(1) Secret Sharing of Local Model: In order to protect the privacy of the filtering results, we implemented the secret-sharing algorithm in our scheme. After training the model using local data and filtering malicious updates, the parameter updates are encrypted rather than being transmitted in plaintext. This ensures that external adversaries cannot infer the intermediate parameter and local data from the ciphertext updates.

(2) Enhanced Robustness via Threshold Secret-Sharing Scheme: Significantly, the use of the threshold secret-sharing scheme ensures robustness even in scenarios involving collusion attacks or node disconnections. The RSU nodes can reconstruct the original secret, providing robustness to the IoV system in unstable communication environments.

Detailed results and information can be found in Table 2.

**Table 2.** Security comparisons of data-sharing schemes.

| Index/Scheme Name | [40] | [41] | [42] | Ours |
|---|:---:|:---:|:---:|:---:|
| Decentralized Management | — | √ | — | √ |
| System Security | — | √ | — | √ |
| Privacy of Data and Models | — | √ | √ | √ |
| Robustness of Global Model | — | √ | √ | √ |
| Nodes Exist Resilient | — | — | — | √ |

## 6. Implementation and Analysis

### 6.1. Experimental Environment Configuration

The experimental hardware configuration includes an Intel Core i7-8700K processor (intel, Santa Clara, CA, USA), GTX 1080T GPU, and 16.0 GB of RAM, running in the Ubuntu 20.04 LTS environment. We utilized FISCO BCOS as the framework, an open-source consortium blockchain platform. Through its provided interface, transactions and smart contracts could be examined. The underlying blockchain and consensus process was implemented using Python (v3.6.3), with smart contracts written in C++. The vehicular federated learning environment was constructed using Python (v3.6.3) and TensorFlow (v2.4.2) for local training and global aggregation. Data collection and blockchain updates were simulated using the OMNET (v5.5.1), integrated with the Simulation of Urban Mobility (SUMO-v1.3.0). To simulate the communication among system nodes, we adopted the Krauss model [43]. During simulation, nodes were observed to move at speeds from 20 to 100 km/h on bidirectional roads within a simulated area of 2.5 km × 2.5 km.

(1) Experimental Model: We adopted the VGG16 image classification model as the global model for federated learning. The VGG16 model consists of eight layers in total, with the first 13 layers being convolutional layers and the last 3 layers being fully connected layers. In the vehicular federated learning environment, the parameters of the VGG16 model were set as follows: learning rate of $1 \times 10^{-3}$ dropout rate of 0.2, and each vehicle performed local training with a batch size of 64, while the RSU performed verification with a batch size of 32.

(2) Datasets: The MNIST dataset [44] consists of 60,000 training samples and 10,000 test samples, each of which is a 28 × 28 grayscale image. This dataset contains 10 classes, representing handwritten digits from 0 to 9, paired with their corresponding labels. The CIFAR-10 dataset [45] is a collection of color images, which comprises 50,000 training samples and 10,000 test samples. Each sample is a 32 × 32 RGB image, and this dataset encompasses 10 types of universal objects, such as "aircraft", "dog", and "car". These two datasets can represent the medium-complexity data collected by in-vehicle local devices, which simulate the image information obtained in real time during driving. They serve as benchmark test data for various FL algorithms designed for mobile edge scenarios.

(3) Node Configuration: As shown in Table 3, we used a configuration of 40 system nodes to simulate data sharing in the IoV. This system scale more precisely replicates the intricacies of the IoV, facilitating the assessment of performance and effectiveness in data-sharing schemes during large-scale deployments. Here, 20% represents the RSUs, while the remaining 80% represents the vehicle nodes. This distribution more precisely mimics the real-world prevalence of RSUs and vehicle nodes, which is a crucial factor in assessing the effects of data sharing and model aggregation among different node types. Moreover, the increased number of vehicles guarantees more frequent local updates, thereby enhancing the reliability and accuracy of the aggregation algorithm. To ensure fairness, equivalent values are configured for the nodes' parameters in these schemes.

(4) Comparative Schemes:

FL [40]: This centralized federated learning approach lacks validation and protection of intermediate parameter updates. It relies on a central server to distribute and aggregate the global model. The accuracy of the global model in this scheme serves as the baseline.

BPFL [41]: This scheme replaces the central aggregator with a consortium chain in the traditional IoV system and applies homomorphic encryption (HE) to protect gradients. Its

global training effect is comparable to that of centralized FL. This scheme enhances the Multi-Krum algorithm to implement gradient detection and filtering, which aligns with our approach. We chose the BPFL as the baseline for comparing with the secret sharing proposed in this paper, allowing us to analyze the computational overhead and global accuracy associated with the encryption and decryption processes.

Madi's FL [42]: This scheme extends the concept of centralized FL and incorporates homomorphic encryption (HE) and verifiable computing (VC) technology, ensuring both privacy protection and verifiability. The experiments with the Fashion MNIST demonstrate the effective accuracy guarantee of the global model.

**Table 3.** Nodes' parameter settings.

| Parameter Name | Parameter Value |
|---|---|
| Number of Task Issuer | 1 |
| Total Number of System Nodes | 40 |
| Total Number of Vehicles | 32 |
| Total Number of RSUs | 8 |

### 6.2. Experimental Results

(1) Model Accuracy: We evaluated the impact of our scheme on the model's accuracy. Subsequently, the average accuracy for each scheme was calculated, and the experimental results are illustrated in Figure 5.

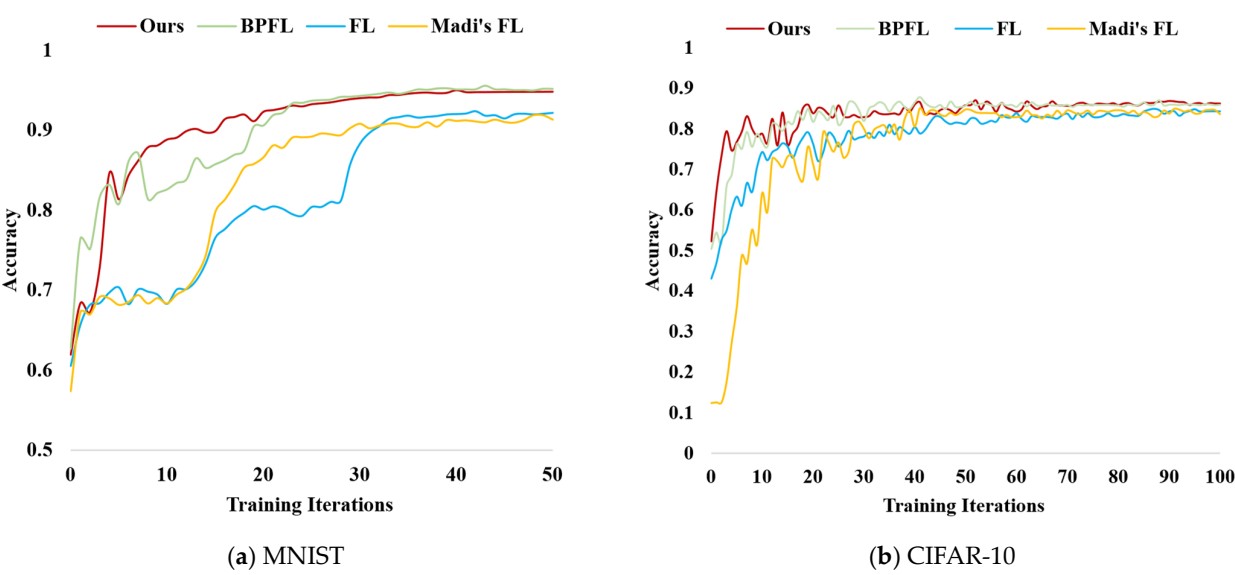

(**a**) MNIST       (**b**) CIFAR-10

**Figure 5.** Model accuracy comparison: Simulations were conducted on the MNIST and CIFAR-10 datasets using four schemes. Among them, FL and Madi's FL schemes do not utilize blockchain, while the BPFL and our proposed scheme are based on blockchain and federated learning. The BPFL uses homomorphic encryption to safeguard intermediate gradients, whereas we utilize a secret-sharing algorithm. The comparison presented in Figure 6 demonstrates the influence of our proposed scheme, which employs blockchain and secret sharing, on the accuracy of the global model.

On the MNIST, these four schemes were trained for 50 rounds, with the BPFL achieving the highest accuracy of 95.56%. Despite our solution not achieving the utmost accuracy, it proved comparable to the BPFL. It is worth noting that our solution displayed a consistently stable accuracy growth curve, resulting in an optimal average accuracy. For the CIFAR-10 dataset, all schemes were trained for 00 training rounds, converging around the 50th epoch. Both our solution and the BPFL scheme realized the highest global model accuracy, with a mere 2.32% disparity.

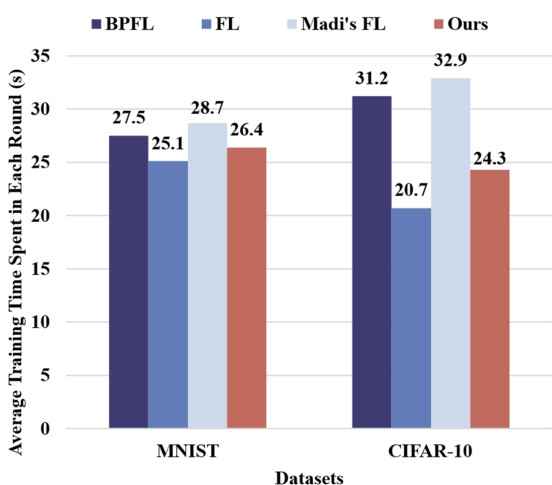

**Figure 6.** Average training time per round comparison.

With an increase in the number of training epochs, the accuracy of the global model steadily improves across all four schemes. Notably, FL and Madi's FL schemes exhibit a consistent upward trend in global model accuracy, whereas the accuracy of the global model in the two blockchain-based federated learning schemes fluctuates during the training process. Moreover, during the initial stages of model training, the blockchain-based schemes achieve higher accuracy for the global model. This can be attributed to the verification of global model accuracy in FL and Madi's FL schemes being conducted by a central aggregator, which utilizes a more concentrated dataset distribution for accuracy evaluation, resulting in a stable upward trend. Both the BPFL scheme and our scheme select intermediate gradients with higher accuracy for aggregation through validation nodes and models, resulting in higher accuracy for the global model in the early stages of training. The validation process for the global model in the BPFL scheme and our scheme involves utilizing local datasets from committee members or validation nodes. This leads to variations in the dataset used for each validation process, resulting in fluctuations in the accuracy of the global model observed in the experimental results.

In conclusion, the BPFL scheme achieves the highest accuracy by conducting a hash function verification and setting parameter retention ratios after completing the training process. Our proposed scheme achieves comparable accuracy to the BPFL but with smoother growth. Before training, we employ a vehicle node selection process based on similarity, which results in optimal average accuracy. As a result, our proposed scheme does not compromise the accuracy of the global model and meets the fundamental requirements of the framework design.

(2) Evaluation of Average Training Time: To evaluate the model training performance of our proposed scheme, we conducted experiments to measure the time required for each round, using the MNIST dataset for 50 rounds of training and the CIFAR-10 dataset for 100 rounds. Subsequently, the average time spent in each training round was calculated, with the duration taken for a single entity to complete its computational task recorded to measure the running time. Both the RSU and vehicles executed computational work in parallel. The experimental results are illustrated in Figure 6.

Figure 6 reveals the minimal differences among the four schemes when applied to the MNIST dataset. Specifically, the average time consumed by training in our scheme was 1.3 s higher than the original FL, accounting for 5.18%. Moreover, it was 1.1 s lower than the BPFL, representing a decrease of 4%, and 2.3 s lower than the Madi's FL, accounting for 8.01%. For the CIFAR-10 dataset, our proposed scheme's training time increased by 3.6 s compared to the original FL, accounting for 17.40%. However, it decreased by 6.9 s compared to the BPFL, accounting for 22.12%. And it decreased by 8.6 s compared to the Madi's FL, accounting for 26.14%. The time consumption gap between our proposed scheme and the original FL remained at 20%. Although the overall time consumption was

higher than that of the original FL, the utilization of the secret sharing and the malicious model filtering, as presented in this paper, may have an impact on efficiency. Moreover, the training time in our scheme was significantly lower than that of both the BPFL and Madi's FL, which employed homomorphic encryption, with the difference reaching up to 26%.

Our research adopted secret sharing over homomorphic encryption, providing a more efficient alternative that reduces computing overhead while ensuring the privacy of the global model. Therefore, our scheme is suited for the IoV scenarios characterized by limited computing power and constrained network communication.

(3) Evaluating Blockchain Consensus: to evaluate the performance of our consensus mechanism, we initially compared the dBFT with several mainstream mechanisms and further validated the advantages of our proposed mechanism, as shown in Table 4 and Figure 7.

**Table 4.** Performance comparison of the consensuses.

| Indicator | PoW | PoS | DPoS | PBFT | dBFT |
|---|---|---|---|---|---|
| Byzantine-fault Tolerance | √ | √ | √ | √ | √ |
| Higher Fairness | √ | √ | — | √ | √ |
| Eventual Consistency | — | — | — | — | √ |
| Low Computing Cost | — | √ | √ | √ | √ |
| Low Transaction Latency | — | — | — | √ | √ |

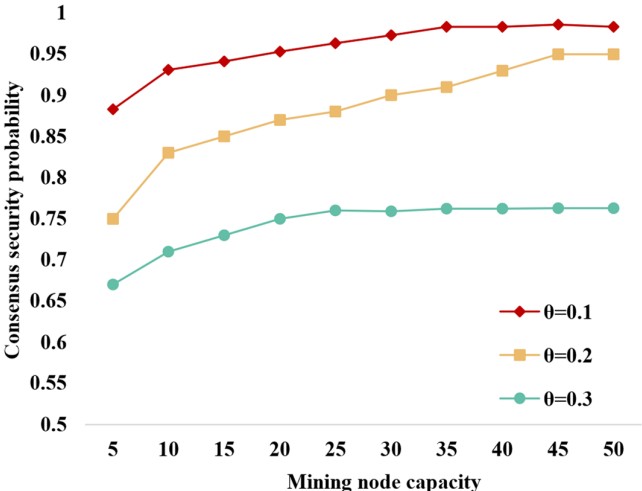

**Figure 7.** Evaluating Blockchain Consensus: Simulations were conducted using the MNIST dataset with 50 mining nodes.

In terms of fairness, the dBFT randomly selects the leader nodes, which is more equitable than the PoS, DPoS, and traditional PBFT, which select leader nodes in a sequential manner. This allows every node to have an equal chance of being selected. Fairness plays a crucial role for the participating nodes in the training process discussed in this paper. Here, each RSU can be chosen as a miner node with accounting rights, ensuring their computing resources are put to meaningful use. In terms of computational consumption, unlike POW, dBFT allows nodes on the blockchain to obtain accounting rights without requiring a significant number of computational resources. This results in higher block generation efficiency by selecting miner nodes to achieve consensus. Once consensus is reached, the Block becomes irreversible and there is no possibility of forks. Compared to the PBFT, the dBFT does not require verification from all nodes on the blockchain, leading to a shorter communication time.

Considering that the Byzantine-fault tolerance rate of the dBFT is one-third, the miner nodes are categorized as positive miners and negative miners. We define the number of negative miners as $N_{neg}$; only when $N_{neg}$ satisfies the condition $N_{neg} < \frac{(N_{Miner}-1)}{3}$ can a

new Block be successfully verified and recorded on the blockchain, where $N_{Miner}$ represents the number of miner nodes in total. Then, we define the following Formula (16):

$$Neg = \sum_{N_{neg}}^{q=0} \binom{N_{Miner}}{q} \theta^q (1-\theta)^{N_{Miner}-q} \tag{16}$$

Here, the parameter $\theta$ represents the probability of becoming a negative miner, with $\theta \in [0.1, 0.3]$ representing the security probability of the miner nodes pool. In this experiment, we set the values of $\theta$ to 0.1, 0.2, and 0.3, respectively, as shown in Figure 7.

The security probability of the dBFT process decreases as $\theta$ increases. Additionally, as the size of the miner node pool increases, the security of the system also increases. This is because the number of positive miners participating in block validation increases with the size of the miner node pool. The larger the size of the miner node pool, the more secure the consensus process in this paper becomes. When a sufficient number of miner nodes participate in the dBFT, this ensures reliable validation for generating new blocks.

(4) Validation of the Vehicle Selection Algorithm: To validate the efficacy of the vehicle selection algorithm, we conducted experiments by varying the number of selected vehicle training nodes. We then applied the algorithm to both the MNIST and CIFAR-10 datasets for testing purposes, with a comparison of the model accuracy presented in Figure 8.

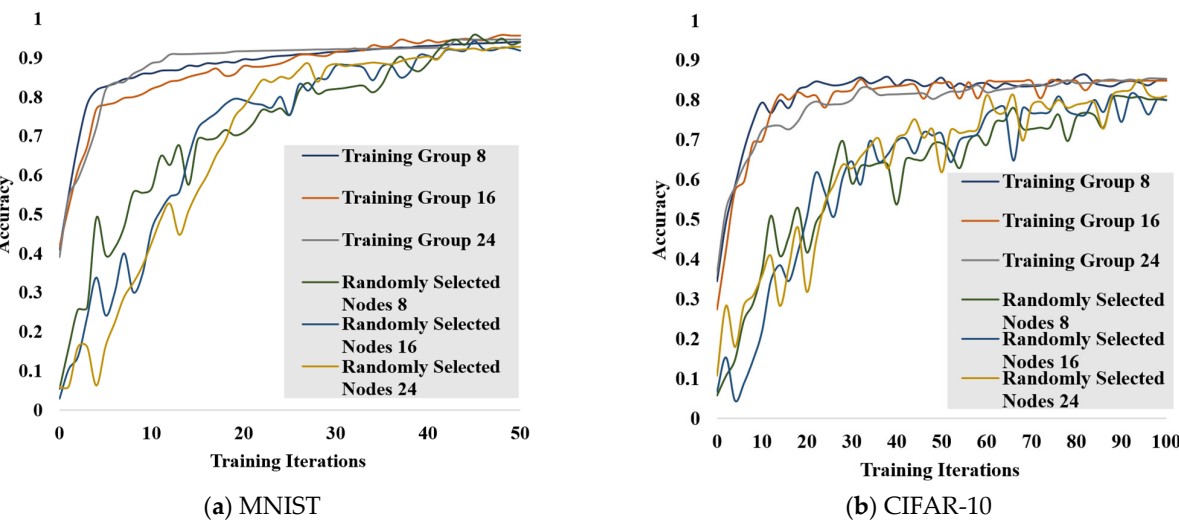

(**a**) MNIST          (**b**) CIFAR-10

**Figure 8.** Vehicle selection algorithm verification: Simulations were performed using the MNIST and CIFAR-10 datasets with 8, 16, and 24 training nodes, respectively. The term "Training Group" refers to the nodes selected by this algorithm, while "Randomly Selected Nodes" represents the training nodes not chosen by this algorithm.

We carried out 50 rounds of training on the MNIST dataset, as depicted in Figure 8a. It is evident that the vehicle selection algorithm was utilized, the global model eventually achieved a stable accuracy. However, the necessary training rounds to achieve this stable accuracy varied. The algorithm-based selection of vehicles, using the concept of Euclidean distance, demonstrated high accuracy from the beginning, with the model quickly converging within approximately 20 rounds of training. Conversely, randomly selected vehicles demonstrated extremely low initial accuracy, and it took longer to reach maximum accuracy, typically requiring about 40 rounds.

Then, we performed 100 rounds of training for the CIFAR-10 dataset, as depicted in Figure 8b. By employing the algorithm-based selection of vehicles, stable convergence was achieved within 35 rounds. In contrast, utilizing randomly selected nodes did not lead to stable convergence even after approximately 65 rounds. Additionally, it can also be seen from Figure 8b that a smaller number of training nodes selected by the algorithm results in faster convergence of the accuracy rate. This outcome can be attributed to the enhanced

correlation between each vehicle and the task issuer, resulting in quicker convergence of the accuracy rate. Hence, the results of the two comparative experiments demonstrate that this selection algorithm significantly enhances the speed and effectiveness of training.

(5) Evaluating Resistance to Poisoning Attacks: To evaluate the robustness of our scheme against poisoning attacks, we conducted experiments using the label inversion attack in [46] to generate poisoning samples. The labels of the training samples were altered while maintaining their original features, with poison sample ratios set at 10%, 20%, and 30% respectively. These manipulated samples were then assigned to designated attackers, and the outcomes were compared against the FL [40] without any poison samples. We then conducted a comparative analysis between our approach and three advanced algorithms specifically designed to mitigate Byzantine attacks.

The experimental results are presented in Figures 9 and 10. For the MNIST dataset, the source label "1" was modified to the target label "8", the source label "2" was modified to "4". For the CIFAR-10 dataset, the source label "dog" was modified to the target label "horse". To minimize the impact of irrelevant labels, binary classifiers were exclusively trained using samples that only contained the source and target labels. Additionally, a random selection of 500 test samples with the source label was conducted to determine the success rate of the attack. The success rate is defined as the percentage of samples for which the source label was predicted as the target label. Subsequently, we partitioned the experimental dataset randomly into local datasets for each vehicle.

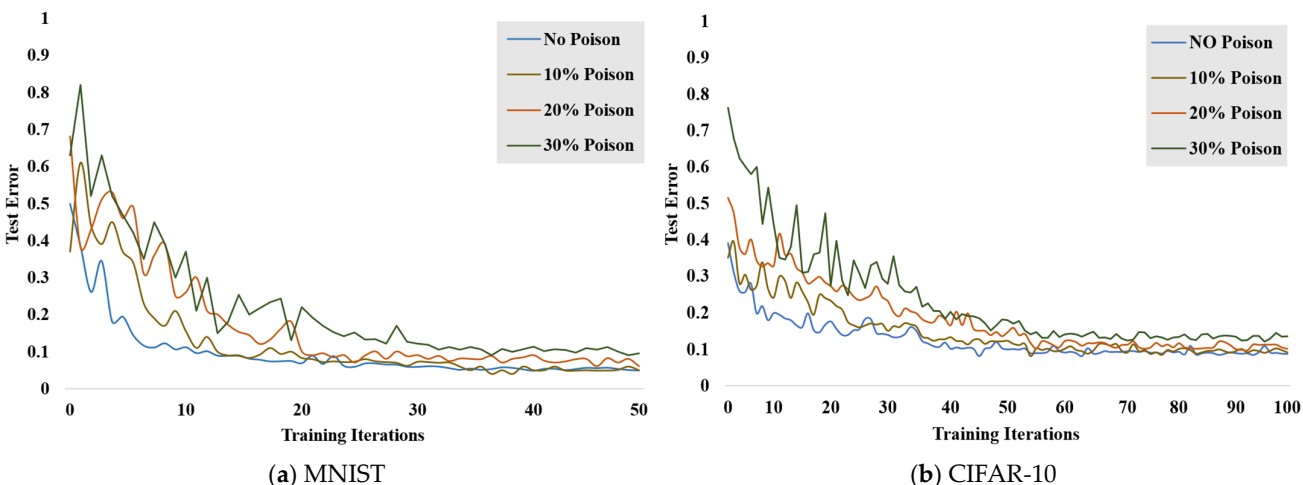

(**a**) MNIST

(**b**) CIFAR-10

**Figure 9.** Test error rate under poisoning attacks: Simulations were performed on the MNIST and CIFAR-10 datasets, where 10%, 20%, and 30% of poisoned samples were introduced, respectively. The results were then compared with the original FL scheme without poisoned samples.

As shown in Figure 9, the poisoning sample ratio was 20%, and the error rate of our approach on both datasets approached 0.1, gradually converging to the level of the FL with no poisoning samples. However, as the poisoning sample ratio increased to 30%, our approach stabilized below 0.16 during subsequent iterations. These findings suggest that our approach can effectively mitigate poisoning attacks with a maximum ratio of 30%.

In order to further validate the performance of our proposed solution, we conducted a comparative analysis by assuming an optimal scenario and comparing it against three other advanced algorithms designed to mitigate Byzantine attacks.

Optimal Scenario: In real-world scenarios where it is impractical to predetermine malicious vehicles, we assume that the RSU has prior knowledge of these malicious nodes. This enables the RSU to autonomously filter out updates uploaded by malicious vehicles, ensuring uninterrupted model training and ultimately resulting in an optimal algorithm outcome. The optimal scenario is employed to assess the accuracy rate of our solution in mitigating poisoning attacks.

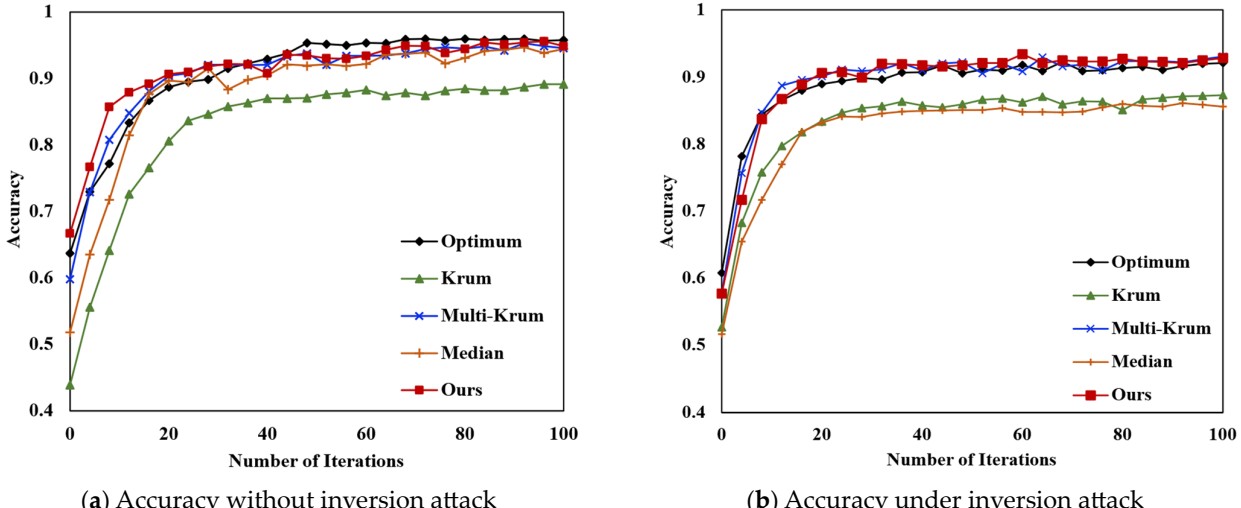

(**a**) Accuracy without inversion attack         (**b**) Accuracy under inversion attack

**Figure 10.** Accuracy under inversion attack: simulations were performed on the CIFAR-10 dataset.

Krum [36]: This algorithm is utilized in distributed machine learning to evaluate the similarity of gradient vectors based on the Euclidean distance. This approach effectively eliminates vectors that deviate significantly, thereby eliminating harmful updates. The calculation steps are as follows:

1. Calculate Euclidean distance: calculate the Euclidean distance $d_{ij}$ between each user-uploaded gradient $LU_i^t$ and the gradient $LU_j^t$ of the other user, $d_{ij} = LU_i^t - LU_j^t$.
2. Select the minimum distance: Identify the set of vectors closest to the vectors of the other $n-2f-2$ users for each user i, where f represents the number of Byzantine nodes. Denote this set of minimum distances as $Dis_i$, $Dis_i = \left\{ d_{i1}, d_{i2}, \cdots, d_{i(n-2f-2)} \right\}$.
3. Calculate the score: evaluate the quality score for each gradient vector by summing the distances in its corresponding set of minimum distances, $Dis_i$, $s(i) = \sum_{j \in D_i} d_{ij}$
4. Choose the best gradient: select the gradient vector with the lowest quality score as the legitimate update for aggregation, $LU_{best}^t = \text{argmin}_i s(i)$.

Multi-Krum [37]: The Multi-Krum algorithm executes the Krum algorithm multiple times, with the calculation steps as follows:

1. For n users, where f is the number of Byzantine nodes (satisfying $n-2f-2$).
2. For each user i, perform the Krum to compute the score $s(i)$ for its gradient vector.
3. Choose the gradient $LU_{best}^t$ with the lowest score as legitimate for aggregation.

Median: The dimensional median algorithm is employed for calculating the global gradient by computing the median along each dimension:

$$\Delta GM_j = \text{median} \left\{ (LU_i^t)_j, \cdots, (LU_k^t)_j \right\}$$

Here, $\Delta GM_j$ represents the j-th dimension of the global model $\Delta GM$, indicating the median value of the local updates contributed by all nodes in the j-th dimension; $(LU_k^t)_j$ represents the j-th dimension of the local update $LU_k^t$; and the function median{·} is utilized to calculate the median value within a set or sequence of numbers.

Additionally, we conducted a comparative analysis of our approach with three other advanced algorithms designed to mitigate Byzantine attacks, using the optimal scenario as the baseline. In Figure 10a, we examined a scenario without any Byzantine nodes, ensuring that each vehicle consistently shared accurate updates with the RSU throughout each iteration. It is evident that both our approach and Multi-Krum showcased promising aggregation performance, attaining an accuracy close to 96% and gradually converging toward near-optimal levels. In contrast, the Krum and Median exhibited poor performance,

deviating significantly from the optimal level. The experimental results under inversion attack are presented in Figure 10b, revealing that our approach and the Multi-Krum showed robust aggregation performance, maintaining an accuracy of over 91% and approaching the optimal accuracy, while the Krum and Median performed poorly, achieving an accuracy of approximately 84%.

In summary, our solution exhibits exceptional resilience when faced with poisoning attacks at a rate of 30%. Our approach maintains a high level of accuracy when compared to the optimal algorithm and three other advanced Byzantine attack mitigation techniques. Consequently, our solution successfully eliminates harmful parameter updates while meeting the fundamental requirements of the framework design.

(6) Analysis of Computational Complexity: We compared our approach with the three advanced algorithms to evaluate its computational complexity in combating poisoning attacks. Table 5 presents the parameters (Params), floating-point operations (FLOPs), and inference time for each algorithm.

**Table 5.** Computational complexity.

| Methods | Params ($10^6$) | FLOPs ($10^6$) | Infer (ms) |
| --- | --- | --- | --- |
| Krum | 0.28 | 132.73 | 5.08 |
| Multi-Krum | 0.47 | 237.58 | 15.13 |
| Median | 9.34 | 94.54 | 3.21 |
| Ours | 0.39 | 82.27 | 8.91 |

Compared to the Multi-Krum algorithm, our algorithm exhibited lower computational complexity and faster inference speed. This is attributed to the introduction of the vehicle similarity calculation at the initial stage, allowing for the selection of vehicle nodes highly compatible with the data provided by the task issuer through Euclidean distance filtering. Additionally, we implemented an evaluation mechanism for the RSU contribution and vehicle credibility. The calculation of the RSU contribution is based on the precision deviation between an RSU and others, indicating the quality advantage of this RSU node relative to others, further ensuring the reliability of collaborative training. This, combined with the enhanced consensus mechanism and on-chain storage, results in faster convergence of our algorithm. Compared to the original Krum algorithm, although our algorithm has slightly more parameters, Krum may result in misjudgments in the presence of only one anomalous gradient, leading to the discarding of updates from honest clients. Therefore, relative to the Krum and Multi-Krum algorithms, our algorithm performs better overall. Compared to the Median algorithm, its faster inference speed is due to its reliance solely on taking the median along dimensions, but it suffers from lower overall model accuracy. In summary, our algorithm not only exhibits outstanding overall performance but also a relatively faster inference speed, making it more suitable for highly dynamic IoV systems with unstable communication.

## 7. Conclusions

This paper introduced the IoV-BDSS, a distributed training scheme for the Internet of Vehicles (IoV), designed to tackle privacy protection and Byzantine attacks in data sharing. Real image data and simulated label reversal attack scenarios were used to evaluate the proposed approach through various performance tests, including accuracy, average time overhead, consensus security assessment, and computational complexity analysis.

The experimental results demonstrate the robust resilience of our approach against 30% poisoning attacks, maintaining a test error rate below 0.16. Compared to similar methods, our approach reduces computational overhead by approximately 26% while preserving accuracy, thereby demonstrating its effectiveness and feasibility in achieving data sharing, privacy protection, and accuracy assurance.



**Author Contributions:** Conceptualization, L.W.; Methodology, C.G.; Software, C.G.; Validation, C.G.; Formal analysis, C.G.; Investigation, C.G.; Resources, C.G.; Data curation, C.G.; Writing—original draft, C.G.; Writing—review & editing, C.G. All authors have read and agreed to the published version of the manuscript.

**Funding:** This work was supported by the Shandong Provincial Key Research and Development Program (2021CXGC010107, 2020CXGC010107), the National Natural Science Foundation of China (62102209), the Shandong Provincial Natural Science Foundation of China (ZR2020KF035), and the New 20 project of higher education of Jinan, China (202228017).

**Data Availability Statement:** Data are contained within the article.

**Conflicts of Interest:** The authors declare no conflict of interest.

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
