# Peer review of "Improving Security in the Internet of Vehicles: A Blockchain-Based Data Sharing Scheme"

_electronics, doi:10.3390/electronics13040714_

Round 1

Reviewer 1 Report

Comments and Suggestions for Authors

<The manuscript may consider accepted, if the following comments are attempted. • There are many grammatical errors. • Quantification of performance gain is missed in the abstract. • Percentage of improvement is missed. Percentage of improvement needs to be revealed after making a comparison with at least 3 state-of-the-art algorithms in the domain considering the same simulation parameters. • Detailed literature review and realistic problem identification can be further elaborated. • The novelty of the manuscript is a bit weak. • Further detailed literature review is needed. • The proposed model is needs to be compared with at least 3 other recently published algorithms in the domain considering the same parameters. • Computational complexity of the proposed algorithm is missed.

Comments on the Quality of English Language

some typo errors and grammatical errors

Author Response

Dear Reviewer,

We thank you for dedicating your time to reviewing our paper titled "Improving Security in the Internet of Vehicles: A Blockchain-based Data Sharing Scheme," with the manuscript number: electronics-2830309. We sincerely appreciate the valuable feedback you have provided and have carefully considered all your comments. We have made the following revisions, highlighted in blue for your reference:

Introduction and Research:

We elaborated on the introduction to provide richer information and enhance coherence. Specifically, we highlighted the significance of the study's background, illustrating the importance of integrating blockchain technology with federated learning in the context of the Internet of Vehicles. This integration not only improves data security, reliability, and trustworthiness but also lowers system expenses and enables swift sharing and exchange of vehicle data. These changes aim to underscore the practical implications of our research and ensure readers have a clearer understanding of the paper's research objectives. Additionally, we revised the research section, focusing on the relevant research status and key real-world problems, particularly addressing privacy protection issues related to gradient detection. We are committed to ensuring the high-quality aggregation of models while enabling the swift sharing and exchange of privacy data in the Internet of Vehicles, thereby strengthening the theoretical foundation and practical significance of our research.

Performance Gains and Innovations:

We thoroughly revised the abstract and conclusion sections, aiming to underscore the practical implications of our approach. Specifically, we highlighted the improvement percentages in addressing poisoning attacks and reducing time overhead compared to similar methods. These changes aim to underscore the practical implications of our approach and provide readers with a more comprehensive understanding of our research outcomes. In the design objectives and security analysis of the paper, we further emphasized the novelty and uniqueness of our approach. We elaborated on the differences between the encryption scheme we adopted and existing solutions and successfully achieved high-precision model aggregation by integrating effective Byzantine fault-tolerant mitigation strategies and consensus mechanisms, demonstrating the advantages of our approach over existing solutions and underscoring its innovativeness and practicality.

Experimental and Analytical Part:

We conducted a more in-depth analysis of the experimental results to ensure they fully support our findings and enhance the credibility of the paper. To ensure the fairness of comparative experiments, we compared our approach with three state-of-the-art Byzantine fault-tolerant mitigation strategies under the same simulation parameters, while also setting an optimal scenario as a reference to comprehensively verify the effectiveness of our approach in addressing Byzantine attacks. Additionally, we provided an analysis of computational complexity to enhance reader comprehension, demonstrating that our approach has smaller computational overhead and faster inference speed, further demonstrating the superiority of our method.

Spelling and Grammar Errors:

We carefully proofread and corrected all spelling and grammar errors, ensuring that the text is free from any errors that may affect clarity and readability.

We sincerely appreciate your thoughtful review and valuable suggestions. We believe that these revisions will further improve the quality of our paper. Once again, thank you for your patience and support.

Best regards,

Chenxi Guan and Lianhai Wang

Reviewer 2 Report

Comments and Suggestions for Authors The paper addresses the security challenges of model aggregation and data sharing that arise during the collaborative training of models by vehicles in the Internet of Vehicles. To overcome these problems the authors propose a framework that is based on blockchain and uses different cryptography schemes to guarantee privacy during data sharing and to ensure the reliability of the training process. This is achieved by the use of Euclidean distance to measure similarity between vehicles and gradients, the employment of a secret sharing algorithm as an alternative to homomorphic encryption and the implementation of a node credibility evaluation system. The system is evaluated and compared to other similar frameworks. The proposed architecture is described thoroughly in sections 3-5 and the evaluation experiments are conducted using very widely used benchmark datasets for computer vision. In the comparison section of the paper the authors compare their approach to other existing systems. However the authors do not discuss adequately the advantages of their proposed methodology, especially in comparison to the similarly performing BPFL approach. A better explanation of the homomorphic encryption employed by BPFL and a more extensive comparison with the proposed secret sharing technique is needed to better show the advantages of the proposed framework, especially the observed large training time gains. More importantly the authors do not discuss adequately why their approach is better security-wise, while BPFL performs better in terms of accuracy. For example, while Table 2 presents a security comparison of the secret sharing schemes employed by the two systems, there is no explanation of the differences presented.   In section 6.1 the selection of node settings is not explained. Section 6.2 is difficult to understand; a table presenting the percentage differences in average training time would be useful. Comments on the Quality of English Language

Some sentences are difficult to understand and need further explanation or have grammatical/syntax errors. They are in lines 71-74, 230-231, 232-233, 522-523.

Author Response

Dear Reviewer,

We thank you for dedicating your time to reviewing our paper titled "Improving Security in the Internet of Vehicles: A Blockchain-based Data Sharing Scheme," with the manuscript number: electronics-2830309. We sincerely appreciate the valuable feedback you have provided and have carefully considered all your comments. We have made the following revisions, highlighted in blue for your reference:

Experimental and Analysis:

We conducted a thorough analysis and revision of our experimental results to ensure their robust support for our conclusions, thereby enhancing the paper's overall credibility. In addition, we compared our approach with three state-of-the-art Byzantine fault-tolerant mitigation algorithms, establishing an optimal scenario as a benchmark to guarantee the fairness of comparative experiments. To comprehensively validate the efficacy of our method in addressing Byzantine attacks, we introduced simulation parameters. Furthermore, we integrated a computational complexity analysis to fortify the manuscript's overall integrity. This supplementary content aims to deepen readers' understanding, facilitating a comprehensive grasp of our method's performance and practicality. Our approach exhibits lower computational overhead and faster inference speed, further substantiating its superiority. The experimental results have been reorganized and presented for clarity and improved understanding, providing robust support for our conclusions.

Innovation and Security Highlights:

We meticulously revised the abstract and conclusion sections, particularly emphasizing the performance gains of our approach. We provided detailed insights into the improvement percentages in countering poisoning attacks and reducing time overhead compared to similar methods. These modifications aim to underscore the real-world effectiveness of our approach, enabling readers to attain a comprehensive understanding of our research outcomes. In the design objectives and security analysis of our paper, we further accentuated the novelty and uniqueness of our approach. Detailed elaboration on the distinctions between our adopted encryption scheme and existing solutions resulted in the successful achievement of high-precision model aggregation through the integration of effective Byzantine fault-tolerant mitigation strategies and consensus mechanisms. These modifications aim to showcase the advantages of our approach over existing solutions and underscore its innovativeness and practicality.

Rationale for Parameter Settings and Section 6.2:

An explanation about node selection was incorporated into Section 6.1 to enhance readers' understanding of our research design. Additionally, Section 6.2 was modified to highlight percentage differences in average training time, further improving reader comprehension.

Spelling and Grammar Errors:

We meticulously reviewed and corrected sentences containing spelling or grammar errors to ensure clarity and understandability. The manuscript is now free of errors that could potentially impact its clarity and readability, particularly in lines 71-74, 230-231, 232-233, and 522-523.

We sincerely appreciate your thoughtful review and valuable suggestions. We believe that these revisions will further improve the quality of our paper. Once again, thank you for your patience and support.

Best regards,

Chenxi Guan and Lianhai Wang

Reviewer 3 Report

Comments and Suggestions for Authors

Summary:

The authors propose IoV-BDSS, a novel data-sharing scheme that integrates blockchain and hybrid privacy technologies to fortify private data in gradient detection.

The effectiveness and feasibility of IoV-BDSS are evaluated through experiments using two real datasets and simulated poisoning attacks to provide a more realistic scenario. The authors conduct multiple performance tests, including accuracy, time overhead, consensus, vehicle selection algorithm and compared with three benchmark schemes.

Comments:

This paper provides a well-investigated study on the data sharing process in the Internet of Vehicle (IoV).

However, this paper has minor issues.

In Figures 2 and 3, the same number are duplicated in the process. Also, since there are typos in the sentences, this paper requires proofreading.

Overall, this paper is well organized and well written to convey the author’s idea and research work.

Comments on the Quality of English Language

In Figures 2 and 3, the same number are duplicated in the process. Also, since there are typos in the sentences, this paper requires proofreading.

Author Response

Dear Reviewer,

We thank you for dedicating your time to reviewing our paper titled "Improving Security in the Internet of Vehicles: A Blockchain-based Data Sharing Scheme," with the manuscript number: electronics-2830309. We sincerely appreciate the valuable feedback you have provided and have carefully considered all your comments. We have made the following revisions, highlighted in blue for your reference:

Graph Issues:

We aim to address concerns regarding the graphs by clarifying that identical numbers in the two graphs correspond to the same step. We have redesigned the charts to provide a clear representation of each step's numbers, eliminating potential confusion stemming from repetition.

Spelling and Grammar Errors:

We conducted a thorough review and revision of the manuscript sentences to enhance clarity and comprehension while rectifying any grammatical errors. Presently, we affirm that the manuscript is free of errors that may compromise its clarity and readability.

We sincerely appreciate your thoughtful review and valuable suggestions. We believe that these revisions will further improve the quality of our paper. Once again, thank you for your patience and support.

Best regards,

Chenxi Guan and Lianhai Wang

Round 2

Reviewer 1 Report

Comments and Suggestions for Authors

The manuscript can be published with its current form.